# The Sense of Number in Fish, with Particular Reference to Its Neurobiological Bases

**DOI:** 10.3390/ani11113072

**Published:** 2021-10-27

**Authors:** Andrea Messina, Davide Potrich, Ilaria Schiona, Valeria Anna Sovrano, Giorgio Vallortigara

**Affiliations:** 1Centre for Mind/Brain Sciences, University of Trento, 38068 Rovereto, Italy; davide.potrich@unitn.it (D.P.); ilaria.schiona@unitn.it (I.S.); valeriaanna.sovrano@unitn.it (V.A.S.); 2Department of Psychology and Cognitive Science, University of Trento, 38068 Rovereto, Italy

**Keywords:** fish cognition, numerosity cognition, quantity discrimination, approximate number system

## Abstract

**Simple Summary:**

The ability to deal with quantity, both discrete (numerosities) and continuous (spatial or temporal extent) developed from an evolutionarily conserved system for approximating numerical magnitude. Non-symbolic number cognition based on an approximate sense of magnitude has been documented in a variety of vertebrate species, including fish. Fish, in particular zebrafish, are widely used as models for the investigation of the genetics and molecular mechanisms of behavior, and thus may be instrumental to development of a neurobiology of number cognition. We review here the behavioural studies that have permitted to identify numerical abilities in fish, and the current status of the research related to the neurobiological bases of these abilities with special reference to zebrafish. Combining behavioural tasks with molecular genetics, molecular biology and confocal microscopy, a role of the retina and optic tectum in the encoding of continuous magnitude in larval zebrafish has been reported, while the thalamus and the dorso-central subdivision of pallium in the encoding of discrete magnitude (number) has been documented in adult zebrafish. Research in fish, in particular zebrafish, may reveal instrumental for identifying and characterizing the molecular signature of neurons involved in quantity discrimination processes of all vertebrates, including humans.

**Abstract:**

It is widely acknowledged that vertebrates can discriminate non-symbolic numerosity using an evolutionarily conserved system dubbed Approximate Number System (ANS). Two main approaches have been used to assess behaviourally numerosity in fish: spontaneous choice tests and operant training procedures. In the first, animals spontaneously choose between sets of biologically-relevant stimuli (e.g., conspecifics, food) differing in quantities (smaller or larger). In the second, animals are trained to associate a numerosity with a reward. Although the ability of fish to discriminate numerosity has been widely documented with these methods, the molecular bases of quantities estimation and ANS are largely unknown. Recently, we combined behavioral tasks with molecular biology assays (e.g c-fos and egr1 and other early genes expression) showing that the thalamus and the caudal region of dorso-central part of the telencephalon seem to be activated upon change in numerousness in visual stimuli. In contrast, the retina and the optic tectum mainly responded to changes in continuous magnitude such as stimulus size. We here provide a review and synthesis of these findings.

## 1. Introduction

The concept of numerosity refers to the cardinality and ordinality of a group of items, and it represents a basic characteristic of the stimuli in the environment [1,2,3]. Widespread research has been done to gather evidence of a non-verbal and non-symbolic capacity for the understanding of the number concept in humans [4,5] as well as in other animal species [6,7,8,9]. It has become apparent that the human capacity to accurately count and perform precise arithmetic arose from a much basic mechanism, likely shared with many animal species, such as mammals [10], amphibians [11], reptiles [12], birds [13,14] and fish [15,16,17].

This system, labeled as “number sense” [18] or “Approximate Number System” (ANS) [18,19,20], is capable of accurately representing numerosities obeying to the Weber’s law, which states that the change of a stimulus that is barely noticeable is a constant ratio of the original stimulus [21]. As to numerosity, this means that the distinguishability of two numerosities decreases as the magnitude of the numbers increases [22,23], the so-called “numerical size effect” [24].

It has been shown that the number sense arises very early during development. In humans, newborns and infants are able to discriminate numerosity of small sets [4,8,25,26]. A few hours old chicks (*Gallus gallus*) are capable of discerning different numerosities [8,13,27,28,29]. Newborns and juvenile fish can be trained to discriminate numerosity [30,31,32].

Since many species display numerical abilities, it has been hypothesized that these abilities guarantee important biological benefits. Numerical skills promote animal’s survival by conferring advantages in food supply [33,34], social interaction [35] and avoiding predation [36,37,38,39].

Given that evolutionarily distant species differ widely in brain organization and complexity, how could they develop similar numerical abilities? This could be either the outcome of common ancestry from which they inherit it, or the outcome of convergent evolutionary processes promoted by similar selective pressures [40,41].

Although an answer to these questions has not yet been provided, subregions of the parietal and prefrontal cortex of humans have been described as involved in numerical skills [42,43,44,45]. Furthermore, neurons with selectivity of response to numerosities have been described in the prefrontal cortex and in the ventral intraparietal area in monkeys [46,47] and in the nidopallium caudolaterale in crows [48,49].

We will review here the behavioural studies that have permitted to identify numerical abilities in fish, and the current status of the research on the neural bases of these abilities.

## 2. Numerical Abilities in Fish

Encoding of numerical information has been shown to provide several advantages to animal species [40,50,51], including those which are more evolutionary distant to humans, such as fish.

For example, fish may join a large shoal when surveying a potentially dangerous location so as to diminish the chance of being predated upon [37,52,53]. Some predator fish species (e.g., wolf-fish pike [*Hoplias malabaricus*] and cichlids [*Crenicichla frenata*]) tend to disregard attacking a single individual and rather focus on large groups of prey [54]. Mosquitofish (*Gambusia holbrooki*) and guppies (*Poecilia reticulata*) can assess the number of females and males present in a group, as part of their reproductive strategies [55,56]; whereas convict cichlids (*Amatitlania nigrofasciata*), a parental care species, are capable of modifying parental activity on the basis of the quantity of their progeny [57].

However, since the numerosity of the stimuli co-vary with other continuous physical variables, the issue has been raised as to the actual mechanisms fish use for discrimination. Rather than discrete numerosities, fish may use the relative magnitude of non-numerical cues (i.e., continuous quantities) such as the cumulative surface area of the stimuli or the density of the elements to be discriminated [58,59,60,61]. Thus, studies that exploit controlled setups and designs are required in order to investigate the processes by which fish are able to process quantity information.

Several techniques have been used to investigate numerical abilities in mammals [62,63,64] and birds [14,27,65,66,67,68]. The most used paradigms were spontaneous choice tests, habituation-dishabituation techniques and operant techniques. Similar methods were used for fish.

## 3. Spontaneous Choice Tests

This procedure exploits the natural ability of fish to discriminate between two groups of biologically relevant stimuli that differ in numerosity, usually food or social companions. The rationale behind this task is that subjects are motivated to choose the larger (or smaller) group since it offers greater survival advantages (higher energy intake or protection).

Compared to other stimuli, fish are mostly attracted by social companions. Several fish species group together (shoal) so to avoid or protect against predation [69]. When shoals have different numbers of companions, fish prefer joining the larger one [37,70,71]. Exploiting this tendency, many studies have investigated quantitative abilities.

Mosquitofish (*Gambusia holbrooki*) appear to be able to discriminate between groups of conspecifics that differ by one unit up to four items (1 vs. 2, 2 vs. 3 and 3 vs. 4, but not 4 vs. 5; [72]). Guppies (*Poecilia reticulata*) show comparable behaviours [15].

Fish are also capable to distinguish between large numerosities (higher than four): swordtails (*Xiphophorus elleri*, [70]), guppies [15] and mosquitofish [72] discriminate two different quantities with a 0.50 ratio (e.g., 8 vs. 16) but not with a 0.67 ratio (e.g., 8 vs. 12), whereas angelfish (*Pterophyllum scalare*) discriminate up to a 0.56 ratio (5 vs. 9; [73]). It has also been shown that fish can exploit quantity discrimination to pick up a more profitable shoal depending on the sex of the composing individuals [58].

In all these tests stimuli are fully visible at the moment of the choice, thus fish could use continuous quantities to choose the larger shoal. Experimental strategies have been devoted to control for the role of continuous quantities, such as the total activity of the moving stimuli. Since many fish species live in a range of temperatures and their activity increases as water temperature would rise, by varying the water temperature between a larger and a smaller shoal, it is possible to balance somewhat the total activity of the two groups. A study showed that zebrafish (*Danio rerio*) preferentially chose the larger shoal over the smaller one (2 vs. 4) when both groups were maintained at the same water temperature [74]. However, any preference disappeared when the temperature of the larger group was decreased, thus diminishing the activity of this shoal. Similarly, mosquitofish choice for the larger shoal seems to be impaired when the overall movement quantity is equalized in small numerical comparisons (i.e., 2 vs. 3; [72]). Thus, it seems that in these cases it is the overall amount of motion rather than number per se that guides fish behaviour.

In order to control for the role of overall cumulative area, shoals comprising bigger or smaller individuals were used. Results showed that mosquitofish did not show any preference for the larger shoal (in 2 vs. 3 and 4 vs. 8 comparisons) when both shoals had the same total surface area [72]. A study conducted on zebrafish, guppy, Chinese crucian carp (*Carassius auratus*) and qingbo (*Spinibarbus sinensis*) also revealed use of the larger cumulative surface area rather than discrete numerical quantities [75].

Density (inter-individual distance) has been studied in angelfish [60]. When fish were tested in a 5 vs. 10 comparison in which the density of the two shoals was made identical, fish exhibited no significant preference for either of the shoals. A control experiment comparing shoals containing an equal number of conspecifics (i.e., 5 vs. 5) but different densities revealed that fish preferred the more dense group suggesting that this continuous physical variable was crucial for angelfish.

A technique to prevent fish from exploiting continuous quantities consists in first presenting two different numerical shoals at the same time and then at test limiting by occlusion the visibility of some items (i.e., one or more stimuli from the larger group are concealed to the testing fish, leaving the same number of stimuli visible in the two shoals). Using this method, zebrafish proved to choose the larger shoal in numerical comparisons involving both small (1 vs. 2 and 2 vs. 3, but not 3 vs. 4) and large numerosities (4 vs. 6, 4 vs. 8 but not 6 vs. 8), with a discriminative accuracy that depended on the ratio between the sets to be discriminated [17]. Similar results were obtained in 27 days post fertilization (dpf) zebrafish larvae in 1 vs. 8 and 1 vs. 3 comparison [32]. Redtail splitfin fish (*Xenotoca eiseni*) tested in small numerical comparisons (1 vs. 2 and 2 vs. 3, but not 3 vs. 4) showed similar performance [16]. Of course, one could argue that the use of continuous physical variables was not apparent here at test but it was coded during initial exposure, and thus maintained in memory.

Another method to control continuous quantities in spontaneous choice tests consists of an “item-by-item presentation” procedure. This paradigm has been used in mammals (e.g., chimpanzees; [76]) and in birds (e.g., chicks; [66]) and consists in a sequential (or simultaneous as control) presentation of elements belonging to each group that prevent subjects a global viewing of the whole contents of the groups. Specifically, in order to solve the task, animals need to keep track of each item to form a representation of the contents of the groups and compare the two quantities.

A study conducted on mosquitofish [77] made use of a paradigm in which each fish stimulus was located in a separate compartment of the tank, and several opaque occluders were inserted so that the subjects could see only one stimulus at a time. Hence, fish were expected to add up the amount of the seen conspecifics on one side, do the same on the other side, and then compare the two quantities in order to pick the preferred shoal. Mosquitofish spent indeed more time nearby the larger shoal in 2 vs. 3 and 4 vs. 8 comparisons.

Besides the use of conspecifics as attractive stimuli, spontaneous choice tests can be used for assessing discriminative judgments between different food quantities. Since more food leads to a better chance of survival, animals are expected to select a larger amount. This method is however less commonly used in fish, due to methodological difficulties in delivering and controlling food because of olfactory cues released in the water. A study in guppies investigated the ability to identify the larger number between two sets of food flakes pasted onto plastic cards. Fish picked up the larger food quantity in 1 vs. 4 and 2 vs. 4 (up to a 0.5 ratio) comparisons, while failing in 2 vs. 3 and 3 vs. 4 [78]. Further experiments showed that guppies paid more attention to cumulative surface area of food items rather than number, showing attraction to the larger food item even when belonging to a set with the smaller overall quantity. In spontaneous foraging tasks, angelfish showed to prefer the numerically larger food set as long as the items were sized identically, with an accuracy that depended on the numerical ratio between the two quantities [79]. However, variables such as the size and density of the food items played an important role [80,81], suggesting that numerical and continuous physical cues may not be considered separately but instead are combined by fish to maximize food intake [82].

## 4. Operant Training Procedures

Spontaneous discrimination takes advantage of ecological and naturalistic setups to investigate quantity discrimination abilities. However, the limitations of this method are apparent, and concern factors such as lack of motivation and difficulty in stimulus control. Discriminative failure may be driven by a lack of motivation, especially when the discrimination involves large numerosities: it is important for animals to maximize the intake strategy when dealing with few items (according to the optimal foraging theory; [33]), but it might not be so relevant when dealing with large numerosities, when both amounts would offer enough energy.

Another issue is related to the difficulty in controlling continuous physical variables that co-vary with numerosity using naturalistic stimuli. Some cues are not easily controllable (e.g., when using social stimuli, the overall movement and the volume of the conspecifics is hard to be taken into account). Besides, the control of some variables does not exclude possible side effects that may influence spontaneous preference, e.g., larger pieces of food may elicit higher attraction [78,80].

Some of these issues may be more easily overwhelmed using artificial and well-controlled stimuli combined with operant procedures. Typically, in training procedures animals are requested to discriminate between different sets of elements with different numerosity by choosing the one associated with a reward (usually food). Differently from spontaneous choice, using discrimination learning procedures it is possible to keep the animal’ motivation high irrespective of the numerosities presented, allowing experimenters to accurately test the actual discriminative limits of the animals’ numerical competence. Agrillo et al. [59] trained mosquitofish (*Gambusia holbrooki*) to discriminate between sets of visual elements (2 vs. 3) and choose the one associated with a reward (i.e., social reward). Mosquitofish proved able to discriminate between the two sets, showing however a drop of performance when either the cumulative surface area or the overall space occupied by the elements was equalized [59]. Similar results were obtained when mosquitofish were trained with large numerosities (higher than four elements; [83]), suggesting that some physical properties are spontaneously used in the learning discrimination process by fish. However, no discrimination impairment was noticed when non-numerical physical cues were simultaneously controlled for during the training [59].

In order to check whether processing numerosity would be more cognitively demanding than processing of continuous quantities (and thus used as a “last resort” strategy, see [84]), mosquitofish were trained in a 2 vs. 3 discrimination by making available either only continuous variables or only numerical information, or both simultaneously. Fish improved their performance when both numerical and physical information positively correlated than when only one of the two information were differing. However, no difference was found between the two latter conditions, suggesting that numerical information is not more cognitively demanding than other types of information [85].

The influence of non-numerical variables has been recently investigated in archerfish (*Toxotes* sp.). In a magnitude discrimination task between two groups of dots differing in number, archerfish showed that choice for sets with more/less dots was mainly modulated by non-numerical magnitudes (i.e., overall surface, overall perimeter, density, convex hull, average diameter) that positively correlated with number. Fish tended to select the group containing the larger non-numerical magnitudes and smaller quantities of dots, choosing the larger group of dots only when it was positively correlating with all non-numerical magnitudes [86].

Despite the large amount of comparative data available in the literature, cross-species comparisons are often difficult because of differences in the methods and in the range of numerical comparisons used. To overcome this problem Agrillo et al. [15] compared the numerical abilities of five teleost fish (guppies, redtail splitfins, angelfish, Siamese fighting fish, and zebrafish) in the same task. Fish were first trained to discriminate different numerosities (for food reward) using numerical sets with a 0.5 numerical ratio (i.e., 5 vs. 10; 6 vs. 12). All the species except angelfish proved then able to generalize to numerosities with a 0.67 ratio (i.e., 8 vs.12) but failed with a 0.75 ratio (i.e., 9 vs. 12). Moreover, fish generalized to novel sets in which the ratio was identical at training (0.5) only when the set size was decreased (i.e., 2 vs. 4), but not increased (i.e., 25 vs. 50). Although the performance among the fish species was similar, the proportion of zebrafish that reached the criterion in the training phase was smaller than in the others. The same pattern was found in a shape discrimination task, suggesting a general learning difficulty rather than a specific deficit in numerical ability in zebrafish [15].

Recent evidence suggests that zebrafish learning performance is strongly influenced by stimulus conspicuosness [87]. Similarly, guppies’ numerical ability is improved when the stimulus saliency is enhanced by the presence of moving targets [88] and is worsened using an automatic conditioning chamber compared to that observed in more naturalistic settings [89]. It is therefore important to take into account that different methods may work well for one species but not for others, and that differences in performance may be related to procedural differences rather than cognitive limitations.

Quantitative abilities have been demonstrated in blind cavefish (*Phreatchthys andruzzii*) [90] trained to discriminate groups of sticks differing in numerosity in a circular thank subdivided in eight equal sectors. The experiment showed that, using the organs of lateral lines, blind cavefish proved able to discriminate between 2 vs. 4 objects when both numerical information and continuous quantities were simultaneously available, with a drop of performance when presented with stimuli controlled for continuous quantities. However, if trained from the beginning only with stimuli controlled for non-numerical quantities, cavefish proved able to learn the discrimination relying solely on numerical information.

Overall, it appears that fish numerical performances are comparable to those of mammals [91], birds [27,92,93], amphibians [11,94], reptiles [12] and invertebrates such as bees [95,96,97], although discrimination accuracy is often lower than in other species such as primates [98,99] and parrots [100]. In these latter cases, however, animals are usually trained for a massive number of trials (thousands of trials), while fish training is usually limited to less than 100 trials. In fish, extensive training can increase numerical performance accuracy as seen in guppy [101] and goldfish [102]. Goldfish can achieve high accuracy levels (>90% correct) when exposed to extensive training (approximately 1200 trials), with performances similar to those of birds [100] and primates [98,99].

The discrimination between two numerical sets of elements may be accomplished using either a relative (choose the larger/smaller) or an absolute (choose a precise number of items) numerical judgement. To disentangle which strategy fish use, guppies were trained to select the larger or smaller of two numerical sets (i.e., 6 vs. 12 elements) and then tested with the trained numerosity against a novel one (i.e., 3 vs. 6 if trained to select the 6; 12 vs. 24 if trained to select the 12). Guppies showed a spontaneous use of the relative numerical judgement (i.e., go for the larger/smaller) rather than an absolute one; despite that, guppies proved to be able to learn also absolute numerical information when specifically trained to do so [103]. Evidence for the use of relative judgements has been found among other vertebrate species (humans: [104]; pigeons: [105], but not in invertebrates where bees seem to spontaneously use absolute rather than relative numerical judgement [95].

Numerical skills are not limited to the cardinal aspect of numerosity; another aspect of numbers is ordinality, namely the ability to identify an item on the basis of its position in a series (e.g., the first, the second…). Evidence of ordinal competence has been reported in mammals [106,107,108], birds [109,110,111] and invertebrates [112,113]. Among fish, guppies proved to be able to learn the position of a feeder in a row (the 3rd in a row of 8). Probe trials excluded the use of absolute spatial information (i.e., the position of the feeder and inter-feeder distance) rather than ordinal numerical one [103]. However, the role of relative spatial distances of the feeder that provided the reward in relation to the entire feeder’s series was not directly investigated by contrasting it with ordinal information. This was done however in zebrafish. Trained to identify the second exit in a series of five equally spaced exits, when at test the absolute spatial cues were placed in conflict with numerical cues, zebrafish performance seemed to rely on numbers. However, zebrafish relied on both numerical and spatial cues when the number of exits was increased (from 5 to 9) and the inter-exit distance was reduced, thus suggesting that relative spatial information (i.e., the spatial position of the correct exit in relation to the overall length of exits’ series) was also taken into account [114]. The mixed-use of ordinal numerical information and relational distances show that as the task became more difficult, redundancy of information was needed to solve it, as it has been found in other species of vertebrates [28].

## 5. From Behavior to Neural Circuits

How does the neural activity in the brain give rise to a specific behavior? Or better, how is it possible to link a specific behavior to neurons and neural circuits associated with it? Although, in recent years, functional magnetic resonance imaging (fMRI) in human and single cell recording in non-human vertebrates led to information gathering cognitive functions and specific brain areas, a systems-level understanding from genetics to behaviors in the same organism is still lacking. The possibility to use a model system in which specific neuronal populations can be genetically manipulated and imaged, providing the possibility to explore the neural correlates of overt behaviors, could represent the ideal strategy to address many questions. Zebrafish represent an excellent vertebrate species for addressing and exploring the neurobiology and genetics of number cognition since this small freshwater cyprinid represents a good compromise between system complexity and experimental simplicity as an animal model to manipulate biology [115,116].

Zebrafish acquired a prominent role as experimental model organisms in biology [117,118,119], moving from classical studies on developmental biology [120,121,122] to molecular genetics with the advent of CRISPR/Cas9 genome editing [123,124] and to high resolution functional circuits studies using confocal microscopy [125,126,127], and optogenetics [128,129]. The transparency of zebrafish larvae and the rapid development of their visual system (from 60 h post fertilization to 9 days of development) provide a unique opportunity to explore visual behavior in response to a large variety of natural and artificial visual stimuli like those used to study continuous and discrete quantities [130,131,132,133,134,135,136,137,138].

## 6. Neural Correlates of a Sense of Continuous Magnitude in Zebrafish

The ability of fish to assess quantity (magnitude) in the continuous domain has been widely studied in zebrafish (see Figure 1 and Table 1 for a summary of the main results and Table A1 for the keywords used in the search strategy). 

Neill and Smith [134] investigated the ability of large populations of zebrafish larval tectal neurons to respond selectively to the size of visual stimuli. They reported that size selectivity was established earlier during development of zebrafish larvae starting from 72 h post fertilization (hpf) for large stimuli and 78 hpf for the small ones, with a perception of magnitude sensitivity and selectivity that improves with the maturation of retinal and tectal dendrites and connectivity (from 84 hpf to 9 dpf; [134]). Following the evidence that vertebrate retina contains distinct populations of retinal ganglion cells sensitive to object size [139,140], Preuss et al. [136] studied distinct populations of tectal neurons involved in the discrimination between small- and large-size objects. Using calcium imaging of retinal ganglion cells (RGC) afferents to optic tectum and artificial stimuli which were previously shown to evoke different swimming patterns [141,142], they showed that RGC afferents and tectal superficial interneurons arborize in distinct retinorecipient layers of the tectal neuropil playing a critical role in object size classification [135]. It was suggested that small-size-selective retinal inputs would arrive at superficial layers of tectal neuropil while large-size-selective ones to deeper layers connecting the size-based categorization of visual targets to the role played by the tectum in approach/avoidance behaviors [135,143]. Barker and Baier [136], combining optogenetics, imaging and single-cell reconstructions, identify specific interneurons in the optic tectum that are tuned to object size, influenced by prey-selective RGCs inputs and thus guiding behavioral choice (approach or avoidance). Finally, Helmbrecht and colleagues [144] extended this research by identifying how the segregation of the outputs generated by the receptive fields is converted into a visual-motor response processed by premotor nuclei located in the hindbrain of zebrafish larva.

Habituation/dishabituation experiments associated with measurements of early gene (IEGs) expression were performed on adult zebrafish by Messina et al. [137]. Animals were first habituated to a set of stimuli (small dots) and then faced (dishabituation) to a similar stimulus with a change in size (threefold increased or decreased). A selective change in the expression of the immediate early genes c-fos and egr-1 in retinal and optic tectum tissues with respect to a group facing the familiar control stimulus was observed [137]. Overall, these findings indicate a conservative role of retina and optic tectum in the elaboration of continuous quantities in embryonic and adult zebrafish.

## 7. Neural Correlates of a Sense of Discrete Magnitude (Number) in Zebrafish

Recently, zebrafish studies have expanded our knowledge about the neural correlates of quantity estimation to discrete quantities (numerosity) (see Figure 2 and Table 1 for a summary of the main results and Table A1 for the keywords used in the search strategy).

Combining a spontaneous habituation/dishabituation paradigm with molecular biology techniques, we explored the major brain regions involved in numerosity discrimination in adult zebrafish [137,138]. Briefly, adult zebrafish were habituated to artificial stimuli (three or nine small red dots that changed in individual size, position and density while maintaining their numerousness and overall surface from trial to trial). During the dishabituation phase, separate groups of fish faced a change in number (nine or three dots with the same overall surface) or different types of change in the stimuli (change in shape or in size) or no change at all (control group). The evaluation of the expression levels of immediate early genes, as specific markers of neural activity, revealed a main role of the thalamus and telencephalon in the elaboration of numerosity [137]. These results are consistent with reports of an activation of thalamic regions in number estimation in human infants using fMRI [145] as well as of an involvement of telencephalic/pallial structures revealed using single cell recording in primates [44,46,47,146,147] and corvids [48,49,148].

Further research aimed to explore more in detail the pallial regions involved in numerosity estimation. This showed a specific activation in the most caudal part of the dorso-central (Dc) area of telencephalon for changes in numerosity, whereas the more rostral part responded to changes in shape [138]. To what extent this area could be considered as an equivalent of the mammalian parietal and prefrontal cortex [42,43,44,45,148] or of the nidopallium caudolaterale of corvids [43,48,49,148,149] is difficult to say. Whether zebrafish Dc is equivalent or homologous to regions in the mammalian and avian pallium [40,150] will require further investigation.

Intriguingly, the increased expression of IEGs in Dc with change from small to large numbers and the opposite trend from large to small numbers suggested that a higher or lower activation of Dc could be associated with motor execution (approach or avoidance) in association with the direction of the change in numerosity [138]. This was in agreement with behavioural measures. These results were also in line with hodological studies reporting that the major descending pathway of fish pallium to optic tectum and medulla oblungata are located in Dc [151,152,153,154,155].

Further research is needed to identify the neural circuits associated with discrete quantity estimation and elaboration in adult and larval zebrafish. In particular, the details of the ascending and descending pathways of Dc with the thalamus and the motoric areas need to be established, and our lab is currently working on this.

## 8. Implications of Neurobiological Research of Number Cognition in Zebrafish

Why is there all this interest in zebrafish’s ability to discriminate continuous or discrete quantities? What are the advantages that an in-depth knowledge of a capacity for quantity discrimination in zebrafish could bring?

Benefits would embrace both fundamental and translational aspects of research. As to the former, the possibility of elucidating the development and the neural connectivity of magnitude/number neurons in wild-type, knock-in and knock-out transgenic lines, as well as the possibility to trace the biological evolution of number sense in vertebrates. As to the latter, research on zebrafish would promise a better understanding and modelling of developmental dyscalculia [156,157].

Zebrafish is a species in which the embryonic development occurs outside of the maternal body. Zebrafish thus offered the unique opportunity to test both embryonic and adult individuals using a smaller or a larger scale version of the same apparatus allowing scientists to explore how the maturation of central nervous system impacts on the emergence of specific cognitive ability [32]. Furthermore, the possibility to combine behavioral approaches, molecular biology techniques and tissue-specific transgenic lines led to the opportunity of studying the brain compartment involved in a specific task and to delineate neural circuits useful to the elaboration of the stimuli. In fact, while the use of transgenic lines has been useful to clarify the contribution and the circuitry connected to the processing of continuous quantities in the embryonic retina and optic tectum of zebrafish [134,135,136,144], molecular biology techniques, such as qPCR and in situ hybridization, proved helpful to identify neural correlates of continuous (retina and optic tectum) and discrete (thalamus and caudal part of *area dorsalis telencephali*) quantity in adult zebrafish brain [137,138].

A deeper understanding of the neural correlates associated with numerical cognition in zebrafish would allow the possibility to address the crucial issue of the evolution of numerical abilities in vertebrates. Are the numerical abilities inherited by a common ancestor or similar selective pressure operated in natural history to develop such abilities in the different species of vertebrates? A limiting aspect here is represented by the difficulties to compare different vertebrates species using exactly the same task. Moreover, the great divergent evolution operated by natural selection in the different vertebrate species makes it difficult a comparison among data collected in one or other species [44,46,47,48,49,146,147,148].

Another advantage is related to the use of zebrafish as animal models to study human developmental dyscalculia. In the last few years, zebrafish became a valid alternative to mouse to model human pathologies [157,158,159] since this fish is easier to genetically manipulate (for example using CRISPR/Cas9 genome editing), to imagine in vivo (because of larval transparency), and to test in large groups of individuals in order to analyze all stage of development in a very short time (from few hours to adulthood). A first screening of gene expression analyses of nine genes (baz1b, fzd9, limk1, tubgcp5, cyfip1, grik1a, robo1, nipa1 and nipa2) associated with human developmental dyscalculia [160,161,162,163] revealed a large expression of all of them in the zebrafish adult pallium and, for five genes (grik1a, robo1, nipa1 and nipa2) an asymmetric distribution between the right and left hemispheres ([157]; see for a general review on brain asymmetry [164] and in fish [165]). The asymmetric distribution of some of the genes associated with human dyscalculia opens the way to the crucial theme that links laterality with number sense and its pathologies [166,167].

## 9. Conclusions

We provided here a state of the art of our current knowledge of the behavior and neurobiology of number sense in fish, and in particular in zebrafish. Although neurobiological studies conducted in different species of vertebrates (mammals, birds and fish) highlighted an involvement of different brain structures in the elaboration of continuous and discrete (numerosity) magnitudes (see for reviews [40,41,150], the neural circuits have not been precisely described as of yet. Research in zebrafish could help to fill this gap allowing to characterize the precise location of the magnitude/number neurons in the brain and to explore the different connectivity associated with the brain regions involved. Moreover, the identification of neurons involved in quantity discrimination processes could lead to the specification of their molecular signatures laying the foundations for comparative molecular studies in other animal species, including humans.

## Figures and Tables

**Figure 1 animals-11-03072-f001:**
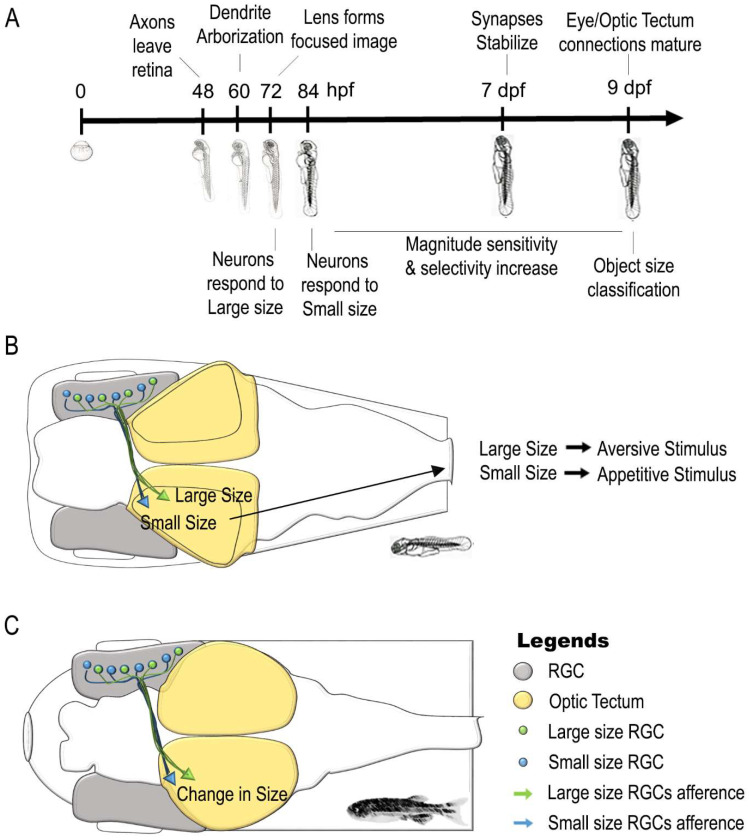
Neural correlates of continuous magnitude estimation in zebrafish. (**A**) Time line of development of continuous magnitude sense in zebrafish embryo and larva. (**B**) Scheme of retinotectal pathways involved in object size discrimination in zebrafish larvae using ethological relevant stimuli. (**C**) The retina and optic tectum are involved in object size classification of visual stimuli in habituation/dishabituation experiments in adult zebrafish brain. See main text and Table 1 for references.

**Figure 2 animals-11-03072-f002:**
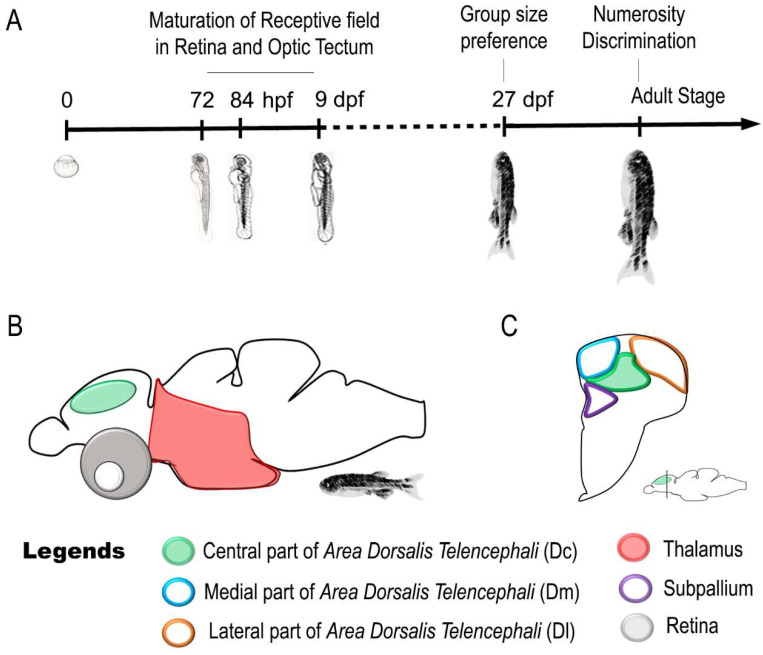
Neural correlates of numerosity cognition in zebrafish. (**A**) Possible timeline of development of number sense in zebrafish. (**B**) Schematic representation of telencephalic and thalamic nuclei activated upon a change in visual numerosity in zebrafish adult brain. (**C**) Molecular biology analyses revealed that the caudal region of central part of the area dorsalis telencephali (Dc) responds to change in numerosity of visual stimuli in adult zebrafish brain.

**Table 1 animals-11-03072-t001:** Summary of the main findings connected to neural correlates of continuous and discrete quantity discrimination in zebrafish.

	Stage	Findings	Literature Data
Sense of Magnitude	72 hpf	Retinal Ganglion Cells (RGCs) respond to Large Size Object	[131,134]
	84 hpf	Retinal Ganglion Cells (RGCs) respond to Small Size Object	[133,134]
	5–8 dpf	Optic tectum contains different population of neurons involved in large and small size object discrimination	[135]
		Retinal Ganglion Cells (RGCs) afferents synapt with Deeper layer of Optic Tectum for Large Size Object	
		Retinal Ganglion Cells (RGCs) afferents synapt with Superficial layer of Optic Tectum for Small Size Object	
	5–8 dpf	Size-based categorization of visual targets and involvement of Optic tectum in approach/avoidance behaviors	[136,143]
	5–7 dpf	Receptive field outputs and visuo-motor response in relation to object size changes	[144]
	9 dpf	Size-based categorization of visual targets similar to adult life	[134]
	Adult	Retina responds to change in size of a visual Stimulus	[137]
		Optic Tectum responds to change in size of a visual Stimulus	
Sense of Number	Adult	Thalamus responds to change in numerosity of a visual Stimulus	[137]
		Telencephalon responds to change in numerosity of a visual Stimulus	
	Adult	The caudal region of the central part of *area dorsalis telencephali* (Dc) responds to change in numerosity of a visual Stimulus	[138]
		Numerosity-based categorization of a visual Stimulus and involvement of Dc in approach/avoidance behaviors	

## Data Availability

Not Applicable.

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
