# Peer review of "The Sense of Number in Fish, with Particular Reference to Its Neurobiological Bases"

_animals, 2021, doi:10.3390/ani11113072_

Round 1
Reviewer 1 Report
In the former half of this manuscript, the authors have given an extensive review on the capacity of many animals to count discrete numbers and to estimate continuous quantity with in-depth reference to the behavioral paradigms used to evaluate these capacities with special emphasis on fish.
In the latter half, they made discussion on the possible neural circuits underlying such abilities in fish. First on the capacity of zebrafish to exhibit different behavioral response (approach or avoidance) to the visual stimuli of different size by referring to the work of Baier’s group which showed how the segregation of the outputs generated by the receptive fields is converted into a visual-motor response processed by premotor nuclei located in the hindbrain of zebrafish larva. Then, the authors discussed about their own works which revealed a main role of the thalamus and telencephalon in the elaboration of numerosity. They showed the increased expression of IEGs in Dc with change from small to large numbers and the opposite trend from large to small numbers, and suggested that a higher or lower activation of Dc could be associated to motor execution (approach or avoidance) in association with the direction of the change in numerosity. Both of these works showed that zebrafish can be a very suitable model animal to understand the neural correlates associated with numerical cognition and the evolution of numerical abilities in vertebrates.
Overall, this manuscript is very exhaustive review of the numerical cognition by animals, and highly intellectually inspiring by pointing out the potential of zebrafish as a promising animal model to elucidate the underlying neural circuit mechanisms. For this reason, I fully support the publication of this manuscript in this journal.
Author Response
Response to Reviewer 1 Comments
In the former half of this manuscript, the authors have given an extensive review on the capacity of many animals to count discrete numbers and to estimate continuous quantity with in-depth reference to the behavioral paradigms used to evaluate these capacities with special emphasis on fish.
In the latter half, they made discussion on the possible neural circuits underlying such abilities in fish. First on the capacity of zebrafish to exhibit different behavioral response (approach or avoidance) to the visual stimuli of different size by referring to the work of Baier’s group which showed how the segregation of the outputs generated by the receptive fields is converted into a visual-motor response processed by premotor nuclei located in the hindbrain of zebrafish larva. Then, the authors discussed about their own works which revealed a main role of the thalamus and telencephalon in the elaboration of numerosity. They showed the increased expression of IEGs in Dc with change from small to large numbers and the opposite trend from large to small numbers, and suggested that a higher or lower activation of Dc could be associated to motor execution (approach or avoidance) in association with the direction of the change in numerosity. Both of these works showed that zebrafish can be a very suitable model animal to understand the neural correlates associated with numerical cognition and the evolution of numerical abilities in vertebrates.
Overall, this manuscript is very exhaustive review of the numerical cognition by animals, and highly intellectually inspiring by pointing out the potential of zebrafish as a promising animal model to elucidate the underlying neural circuit mechanisms. For this reason, I fully support the publication of this manuscript in this journal.
Response 1. Thank you for your kind comments.
Reviewer 2 Report
A revision is required, as the authors did not specify which database/databases and search string(s) were used to collect literature for this review paper. They also need to describe how relevant literature was selected. All this information is relevant and needs to be included.
A minor typing error needs to be corrected. On page 11 the authors write hystory. This must be history.
The manuscript is a review on the capacity in fish to make a distinction between numbers. This was assessed in studies focused on behaviour and neurological analysis. With regard to the behavioural studies the authors discussed various types of studies and which conclusions can be drawn from the results obtained. In my view this manuscript is highly relevant.
A description of a comprehensive literature search is needed for the following reasons:
1) without a comprehensive literature search it is possible that relevant published studies were not included in the review.
Moreover, it is a general requirement for published papers is that it is clearly described how the results were obtained.
One of the reasons for this requirement is that other scientist should be able to repeat published work and thus a section in which the comprehensive literature search is described is essential. Without this description major information is lacking in the manuscript and, therefore, I recommend a major revision.
Hence, the authors need to describe which literature database(s) was (were) used. Which search strings (keywords and combinations thereof) to collect literature were used and which period was covered (e.g. the years 2000-2020 or no time limit). The second step comprises a description of the selection of the collected literature (which criteria were used and who used them).
Please, note that using Google is not a comprehensive literature as in this case the outcome depends on the search history used previously in Google and the location of a computer (e.g. differences between countries can occur). I noticed the typo in the word hystory, which must be written as history.
This is the only typo I noticed. The manuscript is well written.
For both steps the outcome should be presented in a table (e.g. number of published studies found, brief description of criteria used for selection, who did the selection and how many papers were used to prepare the paper) in the manuscript.
A brief text should supplement this table.
Author Response
Response to Reviewer 2 Comments
A revision is required, as the authors did not specify which database/databases and search string(s) were used to collect literature for this review paper. They also need to describe how relevant literature was selected. All this information is relevant and needs to be included.
Response 1. The literature for this review paper was collected using “Web of Science” (https://www.webofscience.com/wos/woscc/basic-search) as browser and using specific search strings reported in the main text. This paper represents, however, a critical review and not a meta-analysis. For this reason, relevant literature was selected by the authors' overlapping lists of papers deriving from each search string. Following the requests of the reviewer, we add a new section to the main text, called “Materials” (pp. 12).
A minor typing error needs to be corrected. On page 11 the authors write hystory. This must be history.
Response 2. We thank the reviewer and we correct the typing error.
The manuscript is a review on the capacity in fish to make a distinction between numbers. This was assessed in studies focused on behaviour and neurological analysis. With regard to the behavioural studies the authors discussed various types of studies and which conclusions can be drawn from the results obtained. In my view this manuscript is highly relevant.
Response 3. Thank you for your kind comment.
A description of a comprehensive literature search is needed for the following reasons:
1) without a comprehensive literature search it is possible that relevant published studies were not included in the review.
Moreover, it is a general requirement for published papers is that it is clearly described how the results were obtained.
One of the reasons for this requirement is that other scientist should be able to repeat published work and thus a section in which the comprehensive literature search is described is essential. Without this description major information is lacking in the manuscript and, therefore, I recommend a major revision.
Hence, the authors need to describe which literature database(s) was (were) used. Which search strings (keywords and combinations thereof) to collect literature were used and which period was covered (e.g. the years 2000-2020 or no time limit). The second step comprises a description of the selection of the collected literature (which criteria were used and who used them).
Response 4. We agree with the reviewer that a description of the criteria used for the collection of the relevant literature, might help the reader to recapitulate and reconstruct described information. For this reason, we add the section “Materials” to the main test (pp. 12).
Please, note that using Google is not a comprehensive literature as in this case the outcome depends on the search history used previously in Google and the location of a computer (e.g. differences between countries can occur).
Response 5. Literature was collected using “Web of Science” as source for our critical review.
I noticed the typo in the word hystory, which must be written as history.
This is the only typo I noticed.
Response 6. We correct the typing error.
The manuscript is well written.
Response 7. Thank you for your kind comment.
For both steps the outcome should be presented in a table (e.g. number of published studies found, brief description of criteria used for selection, who did the selection and how many papers were used to prepare the paper) in the manuscript.
A brief text should supplement this table.
Response 8. We add a table (see Table 2, pp. 12) with the outcome in the main text as suggested by the reviewer. Further details are reported in the section “Materials”.
Reviewer 3 Report
I really enjoyed reading this review. It is well researched and interesting to read. I have a few minor suggestions for consideration by the authors, however I think it is suitable for publication, maybe with very minor changes.
My suggestions are:
Introduction/Abstract: a sentence elabprating a bit more about the significance of numerosity in fish for better understanding human anatomy, neuronal connectivity, and gene function for counting and quantity might be helpful.
In general, consider the wide range of specializations of the readership. Try to use less, or explain preferably with an example, specific scientific terms, so your article is accessible to the entire readership. For example page 5: What does "Fish were facilitated" and "stimulus saliency" mean?
Also on page 5 can you elaborate a bit on the specific test? Which of the sense were used to test quantitative abilities in "blind" cave fish? Smell? Hearing? Touch? etc?And how was the test evaluated? If found this extremely this fascinating, but a bit unsatisfying in content.
The term "rewarded feeder" is confusing or a typo (Page 6)? Did you mean 1) the rewarded individual fish or
2) the rewarding feeder: do you mean the location of a feeder that provided the reward?
Also page 6 and first line page 7 - be cautious with the term "model system". I agree that when you model a specific human condition or disease (diabetes, obesity, alzheimers, etc), zebrafish may be an excellent model for some and not so ideal for others. When referring to models I always like to see the specific "what" of what an organism models. I would suggest not using model unless you include a qualifying example.
Figure 2 (page 10) some areas are difficult to distinguish. e.g., Dm versus Subpallium, DI. Could you use better contrasting colors, and maybe slightly thicker lines?
Author Response
Response to Reviewer 3 Comments
I really enjoyed reading this review. It is well researched and interesting to read. I have a few minor suggestions for consideration by the authors, however I think it is suitable for publication, maybe with very minor changes.
Response 1. Thank you for your kind comments.
My suggestions are:
Introduction/Abstract: a sentence elaborating a bit more about the significance of numerosity in fish for better understanding human anatomy, neuronal connectivity, and gene function for counting and quantity might be helpful.
Response 2. We introduce a sentence in the section “Simple summary” to highlight the significance of numerosity reserch in fish for human studies.
In general, consider the wide range of specializations of the readership. Try to use less, or explain preferably with an example, specific scientific terms, so your article is accessible to the entire readership. For example page 5: What does "Fish were facilitated" and "stimulus saliency" mean?
Response 3. Following the reviewer instruction, we simplify both specific scientific terms using a more accessible language.
Also on page 5 can you elaborate a bit on the specific test? Which of the sense were used to test quantitative abilities in "blind" cave fish? Smell? Hearing? Touch? etc?And how was the test evaluated? If found this extremely this fascinating, but a bit unsatisfying in content.
Response 4. We add in the main text some other details in relation to the task used to assess quantitative ability in “blind” cave fish.
The term "rewarded feeder" is confusing or a typo (Page 6)? Did you mean:
1) the rewarded individual fish or
2) the rewarding feeder: do you mean the location of a feeder that provided the reward?
Response 5. The term “rewarded feeder” refers to the location of the feeder that provided the reward. We specified this is the main text.
Also page 6 and first line page 7 - be cautious with the term "model system". I agree that when you model a specific human condition or disease (diabetes, obesity, alzheimers, etc), zebrafish may be an excellent model for some and not so ideal for others. When referring to models I always like to see the specific "what" of what an organism models. I would suggest not using model unless you include a qualifying example.
Response 6. We agree with the reviewer and modify the term “model system” with “vertebrate species”.
Figure 2 (page 10) some areas are difficult to distinguish. e.g., Dm versus Subpallium, DI. Could you use better contrasting colours, and maybe slightly thicker line
Response 7. We modify figure 2 to better contrast the colours of different brain areas.
Round 2
Reviewer 2 Report
None